# Numerical Simulation of Combustion in 35 t/h Industrial Pulverized Coal Furnace with Burners Arranged on Front Wall

**Jiade Han [1], Lingbo Zhu [1], Yiping Lu [1,*], Yu Mu [1], Azeem Mustafa [2] and Yajun Ge [1]**

[1]   Department of Mechanical and Power Engineering, Harbin University of Science and Technology, Harbin 150080, China; jdhan@hrbust.edu.cn (J.H.); z1543536222@163.com (L.Z.); muyu1239@163.com (Y.M.); ge_yajun@163.com (Y.G.)

[2]   School of Energy Science and Engineering, Harbin Institute of Technology, Harbin 150001, China; azeem.mustafa6@gmail.com

*   Correspondence: luyiping@hrbust.edu.cn

**Abstract:** Coal-fired industrial boilers should operate across a wide range of loads and with a higher reduction of pollutant emission in China. In order to achieve these tasks, a physical model including two swirling burners on the front wall and boiler furnace was established for a 35 *t/h* pulverized coal-fired boiler. Based on Computational Fluid Dynamics (CFD) theory and the commercial software ANSYS Fluent, mathematical modeling was used to simulate the flow and combustion processes under 75% and 60% load operating conditions. The combustion characteristics in the furnace were obtained. The flue gas temperature simulation results were in good agreement with experimental data. The simulation results showed that there was a critical distance $L$ along the direction of the furnace depth (x) and $Hc$ along the direction of the furnace height (y) on the burner axis. When $x < L$, the concentration of NO decreased sharply as the height increased. When $y < Hc$, the NO concentration decreased sharply with an increase in the $y$ coordinate, while increasing dramatically with an area-weighted average gas temperature increase in the swirl combustion zone. This study provides a basis for optimizing the operation of nitrogen-reducing combustion and the improvement of burner structures.

**Keywords:** swirling burner; combustion characteristics; CFD; industrial pulverized coal furnace

## 1. Introduction

At present, with the increasing awareness of environmental protection, the pollutant emissions from coal-fired power stations, industrial boilers and industrial furnaces are widely concerning. When this fuel-burning equipment is used for industrial activities, it is crucial to better understand the combustion characteristics inside the furnace. The distribution characteristics of pollutant emissions (e.g., NO emissions), mechanism of NO formation, and emission reduction technology have become some of the hottest issues worldwide [1,2]. There has been increasing cooperation among countries to improve the performance of coal-fired boilers and reduce emissions [3,4]. Combustion is a complex phenomenon with flow–heat–mass transfer and chemical reactions. With the development of powerful computer hardware and advanced numerical techniques, simulation methods have been adopted to solve the problem of actual boiler combustion. More accurate and reliable results can be obtained [5,6]. Therefore, this kind of numerical simulation can provide theoretical guidance for the modification of burner structure and boiler operation.

In recent years, various forms of burners and graded air supplying methods have been developed in various countries to decrease nitrogen emissions from combustion flames using numerical

techniques together with experimental methods. For example, Hwang et al. [7] performed a full-scale Computational Fluid Dynamics (CFD) simulation of the combustion characteristics of their designed burner and an air staging system for an 870 MW pulverized boiler for low rank coal, and the results of the simulation agreed well with measurements obtained under normal operating conditions, e.g., temperature. Noor et al. [8] studied the influence of primary airflow on coal particle size and coal flow distribution in coal-fired boilers. Silva-Daindrusiak et al. [9], on the other hand, used numerical methods to simulate the three-dimensional flow field in the combustion chamber and heat exchanger of a 160 MW pulverized coal furnace with the objective of identifying factors of inefficiency. The code was built and combined together with the commercial software ANSYS Fluent. Constenla et al. [10] predicted the flow of a reactive gas mixture with pulverized coal combustion occurring in a tangentially fired furnace under actual operating conditions, and shared the experience of ensuring the stable convergence of the equations. Similar to the work mentioned above [10], Srdjan Beloševi et al. [11] simulated the emission of nitrogen oxides and sulfides in a lignite boiler, performed with an in-house developed numerical code. Wang et al. [12] developed a low-volatile coal combustion system with four corners for the burner and two tangential air distribution systems on the four walls, and with the main combustion nozzle arranged on the sidewall near the center of the flame. On this basis, the factors that influence the $NO_X$ generation and the flow field during multi-scale fractional combustion were studied. Sung et al. [13] investigated the combustion performance and $NO_X$ emissions of nontraditional ring-fired furnaces, including models with an additional inner water wall, based on a traditional 500 MW tangentially fired furnace, using commercial CFD code.

Many researchers have contributed to the study of mechanisms to reduce pollutant production. For example, Lisandy et al. [14] established drop-tube furnace (DTF) modeling using one-dimensional numerical simulation for the rapid, good-accuracy prediction of the output gas and unburned carbon (UBC) concentrations. The results showed that the existence of a high CO gas concentration would assist in suppressing the NO formation during the combustion of coal char particles of pulverized coal. The models were crucial for accurate predicted simulation results. Zhang et al. [15] proposed a char combustion order reduction model, successfully implemented into the CFD software, Fluent, and applied to a parallel CFD simulation of a 600 MW boiler. For the efficient utilization of variable quality fuels, a three-dimensional numerical model and experiments were used by Shen et al. [16] to simulate the flow and combustion of binary coal blends under simplified blast furnace conditions, and they found that the chemical interactions between two components in terms of particle temperature and volatile content are responsible for the synergistic effect.

In summary, it can be seen that most of the published studies have carried out experiments using air depth classification technology that changes the burner structure and fractional rate of air supply. Furthermore, numerical simulations of the characteristics of the flow field and combustion processes, as well as reaction mechanism, model and pollutant emission prediction research both have been carried out. However, there have been few studies on the combustion situation of industrial pulverized coal boilers at different loads.

The main objective of the present paper is to present a CFD numerical method for studying the combustion performance in 35 t/h industrial pulverized coal boilers with two swirl burners arranged on the wall. The fluid flow and temperature field of the flue gas near the burner and the distribution characteristics of the combustion products (i.e., NO, $O_2$, CO, $CO_2$ and unburned carbon concentrations) were investigated at two loads to provide basic data for optimizing the combustion operation for nitrogen reduction in industry boilers.

## 2. Physical Model and Meshing

### 2.1. Physical Model

In the present work, we studied the combustion characteristics of a 35 t/h pulverized coal boiler with two swirl burners on the front wall. The combustion characteristics were closely related to the

boiler structure of the chamber, burner and air staging system designed, the operating conditions, and other factors. The burner used by the boiler was mainly composed of an ignition device, central tube, primary air (PA) duct, internal secondary air (SA) duct and external SA duct. Figure 1 shows that the central air (CA) duct was arranged in the PA duct, which was equipped with a conical nozzle. As Figure 1b shows, there were 12 fixed guide vanes uniformly arranged along the circumferential direction in the inner and outer SA ducts, respectively, to achieve the air staging supply of SA of the inner and outer swirls. The CA can effectively regulate the center location of the combustion flame; moreover, it is also a key factor affecting flashback characteristics. An air staging system is also helpful for adjusting the radial velocity profile of the burner exit.

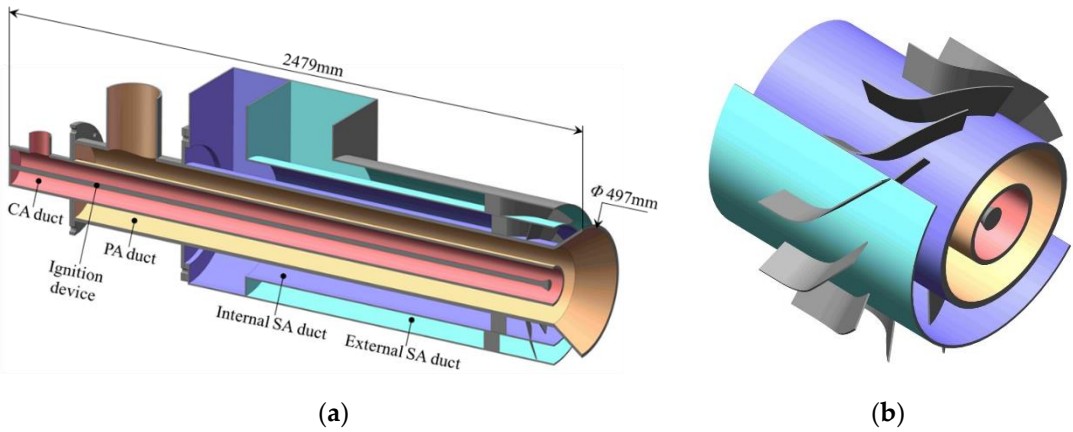

(**a**)　　　　　　　　　　　　　　　　(**b**)

**Figure 1.** Schematic diagram of swirl burner. (**a**) Structure of swirl burner; (**b**) Vane geometry.

The furnace height, width and depth were 13.115, 4.24 and 4.37 m, respectively. The nozzles of the two burners were arranged on the water-cooling wall surface of the front wall, the horizontal spacing of the two burner central axes was 1.7 m, and the distance from the burner central axis to the furnace bottom was 4.35 m, as shown in Figure 2a. Figure 2b shows a local view of the part connecting the burners with the furnace front wall, and we can see how the SA was discharged into the furnace through the fixed vanes. To undertake the simulation, the commercial software SOLIDWORKS was used to establish the overall computational domain of the two swirling burners and furnace.

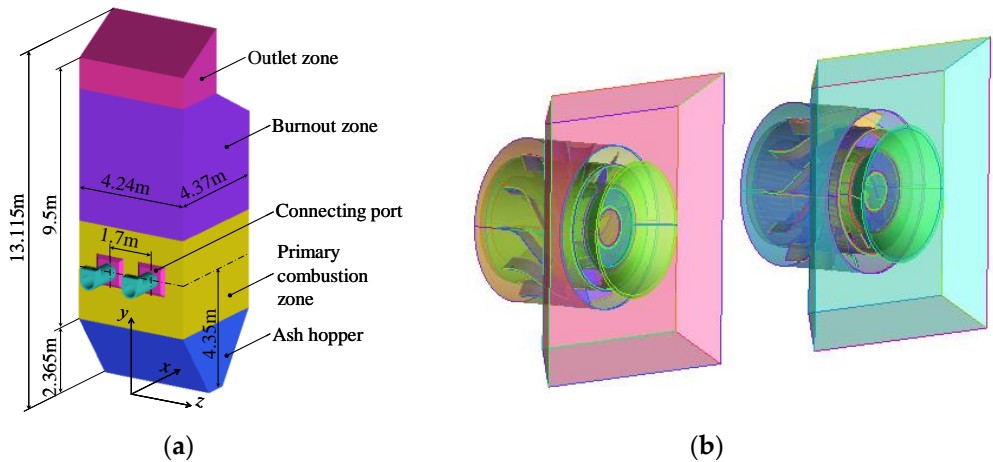

(**a**)　　　　　　　　　　　　　　　　(**b**)

**Figure 2.** Computational domain. (**a**) Computational domain; (**b**) Local view of the part connecting the burners with the furnace front wall.

## 2.2. Meshing

A structured mesh was generated by using the commercial software ICEM, and after several trial calculations, the corresponding mesh was refined for the burner and the primary combustion zone with a large magnitude of the gradient of physical quantities. An initial mesh with about 1.40 million cells was first created in the computational domain. The number of mesh cells was then increased to 2.3, 4.0 million, respectively. The Grid Convergence Index (GCI) was used to quantify the grid independence [17]. The $GCI_{12}$ for fine and medium grids was 1.21%. The $GCI_{23}$ for medium and coarse grids was 3.09%. The value of $GCI_{23}/(r^p GCI_{12})$ was 1.015, which was approximately 1 and indicates that the solutions were well within the asymptotic range of convergence. The mesh-independence test was verified by comparing the temperature of the monitoring point (y = 10.5 m, x = 0.21 m) simulated on three grids and calculated via Richardson extrapolation; see Figure 3. Finally, the numerical results showed that the number of mesh cells was approximately 2.3 million, as shown in Figure 4.

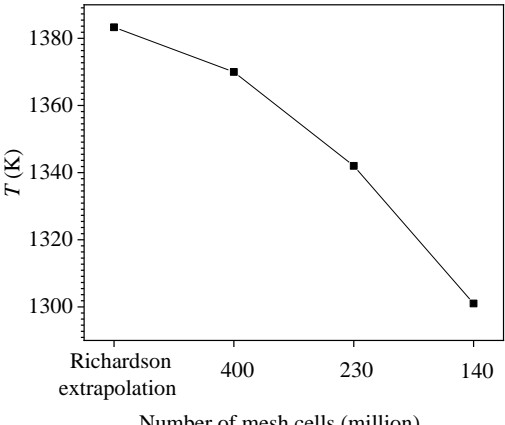

**Figure 3.** The temperature of a monitoring point simulated on three grids and calculated via Richardson extrapolation.

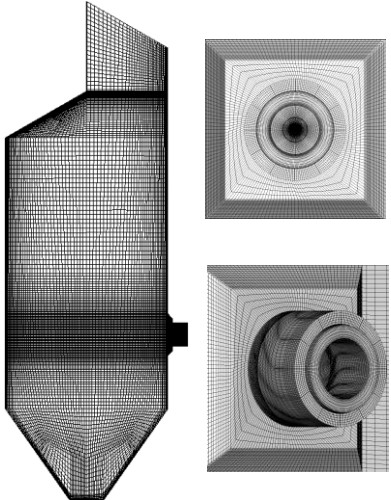

**Figure 4.** Meshing of computational domain.

## 3. Mathematical Model and Solution Conditions

In this study, the commercial software Fluent 16.0 was used to predict the behavior of reacting gases and the coal combustion in the furnace; the CFD parameters had been widely used and were referenced in many coal combustion simulation studies. Due to the inside of the furnace having a

strongly swirling flow, the $k - \varepsilon$ realizable model seemed to be a suitable model for calculating the turbulent viscosity, $\mu_t$ and other parameters of gas turbulence [6,10]. During the combustion of coal particles, several processes occur, which was difficult to model; a non-premixed combustion model was used, and a stochastic tracking model was applied to analyze pulverized-coal flows, while calculations of gas/particle two-phase coupling employed the particle-source-in-cell method [18]. The release rate for the volatiles was expressed with the two competing rates model; Arrhenius equations are used for predicting this chemical reaction rate. The gas-phase turbulent combustion of volatiles was modeled by employing probability density function (PDF) theory; char combustion was expressed with a diffusion/kinetics model. The P-1 model was used to account for the radiative heat transfer effect. Detailed equations for mass, momentum, energy and species transport were provided by [6,19]. These models provided good approximate solutions for a full-scale boiler simulation.

Here, the specific models and equations in Fluent that were used, such as for the mass, momentum, energy, chemical species etc., are listed as follows:

$$\frac{\partial \rho}{\partial t} + \frac{\partial (\rho u_i)}{\partial x_i} = 0, \tag{1}$$

$$\frac{\partial \rho u_i}{\partial t} + \frac{\partial \rho u_i u_j}{\partial x_j} = \frac{\partial \tau_{ij}}{\partial x_j} - \frac{\partial p}{\partial x_i} + \rho f_i, \tag{2}$$

$$\frac{\partial \rho h}{\partial t} + \frac{\partial \rho u_i h}{\partial x_i} = \frac{\partial q_i^{res}}{\partial x_j} + \frac{\partial p}{\partial t} + u_i \frac{\partial p}{\partial x_i} + \tau_{ij} \frac{\partial p}{\partial t} + S_h, \tag{3}$$

$$\frac{\partial \rho m_l}{\partial t} + \frac{\partial (\rho m_l u_i + j_i)}{\partial x_i} = S_l, \tag{4}$$

$$\frac{\partial}{\partial t}(\rho k) + \frac{\partial}{\partial x_i}(\rho k u_i) = \frac{\partial}{\partial x_j}\left[\left(\mu + \frac{\mu_t}{\sigma_k}\right)\frac{\partial k}{\partial x_j}\right] + G_k + G_b - \rho\varepsilon - Y_M + S_k, \tag{5}$$

$$\frac{\partial}{\partial t}(\rho\varepsilon) + \frac{\partial}{\partial x_i}(\rho\varepsilon u_i) = \frac{\partial}{\partial x_j}\left[\left(\mu + \frac{\mu_t}{\sigma_\varepsilon}\right)\frac{\partial\varepsilon}{\partial x_j}\right] + \rho C_1 S_\varepsilon - \rho C_2 \frac{\varepsilon^2}{k + \sqrt{v\varepsilon}} + C_{1\varepsilon}\frac{\varepsilon}{k}C_{3\varepsilon}G_b + S_\varepsilon, \tag{6}$$

$$\frac{du_p}{dt} = F_D(u - u_p) + \frac{g_x(\rho_p - \rho)}{\rho_p} + F_x,$$

$$F_D = \frac{18\mu}{\rho_P d_P^2}\frac{C_D R_e}{24}, C_D = a_1 + \frac{a_2}{R_e} + \frac{a_3}{R_e}, \tag{7}$$

$$q_r = -\frac{1}{3(\alpha + \sigma_s) - C\sigma_s}\nabla G = \Gamma\nabla G,$$

$$\nabla(\Gamma\nabla G) - \alpha G + 4\pi\sigma T^4 = S_G, \Gamma = -\frac{1}{3(\alpha + \sigma_s) - C\sigma_s}, \tag{8}$$

$$k_n = A_n exp(-E/RT_p), n = 1,2,$$

$$dV/dt = dV_1/d\tau + dV_2/d\tau = (\alpha_1 A_1 + \alpha_2 A_2)W, \tag{9}$$

$$\frac{dm_p}{dt} = -\pi d_p^2 p_{ox}\frac{D_0 R}{D_0 + R},$$

$$D_0 = C_1\frac{(T_P - T_\infty)}{d_p}, R = C_2 e^{-(E/RT_p)}, \tag{10}$$

where $x_{i,j}$ = Cartesian coordinates; $t$ = time; $u_{i,j}$ = the velocity in directions $i$ and $j$, respectively; $P$ = the pressure; $\rho/\rho_p$ = the gas/partial density; $\tau_{i,j}$ = the viscous stress tensor; and $f_i$ = the external force. In the law of conservation of energy formulation 3, $h$ = the specific enthalpy, $q_i^{res}$ is associated with energy transfer by the conduction and diffusion flux of matter, and $S_h$ = a source of energy due to

chemical reactions and radiative heat transfer [6]. $m_l$ = the mass concentration of the components $l$, $j_i$ = a weight average flow in the $_i$-th direction, $S_l$ = components of source term $l$; $k$, $\varepsilon$ = the turbulence kinetic energy and its rate of dissipation,; more details ae given in [19]. Only steady state was modeled and analyzed in this study.

The coal combustion simulation required that values be assigned to many input parameters. The coal properties are listed in Table 1. The particle size of the coal at each nozzle follows the Rosin–Rammler distribution; the particle size here was 75–100 μm. The test results showed that the end of the flame in the furnace was closer to the back wall at rated load, and the boiler mainly worked when the peak load was adjusted to under 75% or 60% for low-load operating conditions in the context of safety. Under different load conditions, the boundary conditions for each burner are listed in Table 2. The flame length of the flue gas in the furnace was relatively within the safe range, where the PA volume accounted for 20% of the total air volume, and the SA volume of both the internal and external air volume accounted for 40% of the total air volume under the current operating load, respectively.

**Table 1.** Coal properties.

| Proximate Analysis (As Received) (wt.%) | |
| --- | --- |
| Moisture | 3.40 |
| Volatile matter | 31.58 |
| Fixed carbon | 56.64 |
| Ash | 8.38 |
| **Ultimate Analysis (Dry Basis) (wt.%)** | |
| C | 71.71 |
| H | 4.30 |
| N | 1.01 |
| O | 10.90 |
| S | 0.30 |
| Low heating value | 27.95MJ/kg |

**Table 2.** Settings of the burner inlet parameters.

| Load (%) | Boiler Capacity (t/h) | Mass Flow Rates of Coal in a Single Burner (kg/s) | Temperature of PA Stream and Pulverized Coal (°C) | Temperature of Internal and External SA Streams (°C) |
| --- | --- | --- | --- | --- |
| 75 | 26.25 | 3.96 | 140 | 250 |
| 60 | 21 | 3.17 | 140 | 250 |

The water tubes in the furnace were simplified as walls; a "no-slip", "no-turbulence" and "no-mass flow" boundary condition was employed along the wall for the gas phase. Because the temperature difference of the wall was not large, the wall temperature could be assumed as uniform and was set as 254 °C, higher than the saturation temperature (204 °C) of the water and steam at the working pressure (16 MPa) of the boiler, because the temperature difference of the wall was small. The thermal radiation coefficient set for the wall regions was 0.6. The outlet boundary was the flue gas passage at the lateral wall near the top of the boiler, where the mean static pressure was −50 Pa.

The mass, pressure, momentum, chemical species and energy equations were each discretized using a second-order upwind scheme; a separation and implicit solution scheme based on a pressure solver was adopted. In solving algebraic equations, the SIMPLE algorithm for the pressure correction was applied to couple the velocity and pressure fields [20,21]. Many important scientific and engineering applications involving turbulent flows require the direct integration of the modeled turbulent equations to a surface boundary. Turbulent flows involve boundary layer separation or complex alterations of the surface transport properties [22]. The near-wall modeling significantly impacts the fidelity of numerical solutions, inasmuch as walls are the main source of mean velocity and turbulence. Here,

the standard wall function approach was used for near-wall treatment; also, the local mesh refinement of the boundary layer near the wall surface was performed by adaption to make the dimensionless distance $y^+$ meet the standard wall function requirements ($y^+ > 30$) in the globe range. At section $y = 4.35$, the $y^+$ value is 50.9, for example.

Before solving the combustion process, the simulation of the cold flow field was conducted, and the convergence criterion adopted was the RMS (Root Mean Square of the residual values). After the convergence of the flow field iteration, energy, radiation, species and discrete-phase model (DPM) equations were then activated. The number of iterations was set until the solution with successive under-relaxation satisfied the pre-specified tolerance. Some settings and parameters of the Non-Premixed combustion and DPM models are listed in Table 3.

**Table 3.** Settings and parameters of Non-Premixed combustion and discrete-phase model (DPM) models.

| Non-Premixed Combustion Model | Chemistry State Relation | Chemistry Energy Treatment | Chemistry Stream Option | Operating Pressure | PDF Opinion |
|---|---|---|---|---|---|
| Settings | Chemistry equilibrium | Non-adiabatic | Empirical stream | 101,325Pa | Inlet Diffusion |
| **Discrete Phase Model** | **DPM Iteration Interval** | **Max. Number of Steps** | **Specify Length Scale Length Scale** | **Turbulent Dispersion Discrete** | **Number of Tries** |
| Settings | 20 | 5000 | 0.005 | Random walk model | 10 |

Finally, the NO*x* was computed using the solution obtained with the previous steps. NO*x* formation includes thermal NO*x* and fuel NO*x* but hardly any prompt NO*x*. In this study, the formation of prompt NO*x* was neglected in calculations, and only NO production was taken into account because the NO*x* emitted into the atmosphere from combusting fuels consists mostly of NO. The concentration of thermal NO*x* was calculated using the extended Zeldovich mechanism (specifically, $N_2 + O \rightarrow NO + N$, $N + O_2 \rightarrow NO + O$, $N + OH \rightarrow NO + H$) [23]. The fuel NO*x* concentration was calculated using De Soete's model [23,24].

## 4. Results and Discussion

### 4.1. Analysis of the Fluid Flow and Temperature Field of the Burners

In order to not only reflect the temperature value of the high-temperature flue gas generated by the combustion reaction near the furnace burner region, but also show the characteristics of the vortex flow field more clearly, Figure 5 shows the velocity vector and gas temperature profile of the swirl burner in the longitudinal section of the primary zone where $z = 0.85$ m (a), center axes of horizontal section where $y = 4.35$ m (b), and the gas temperature contour at different y planes of the furnace under two kinds of loads (c).

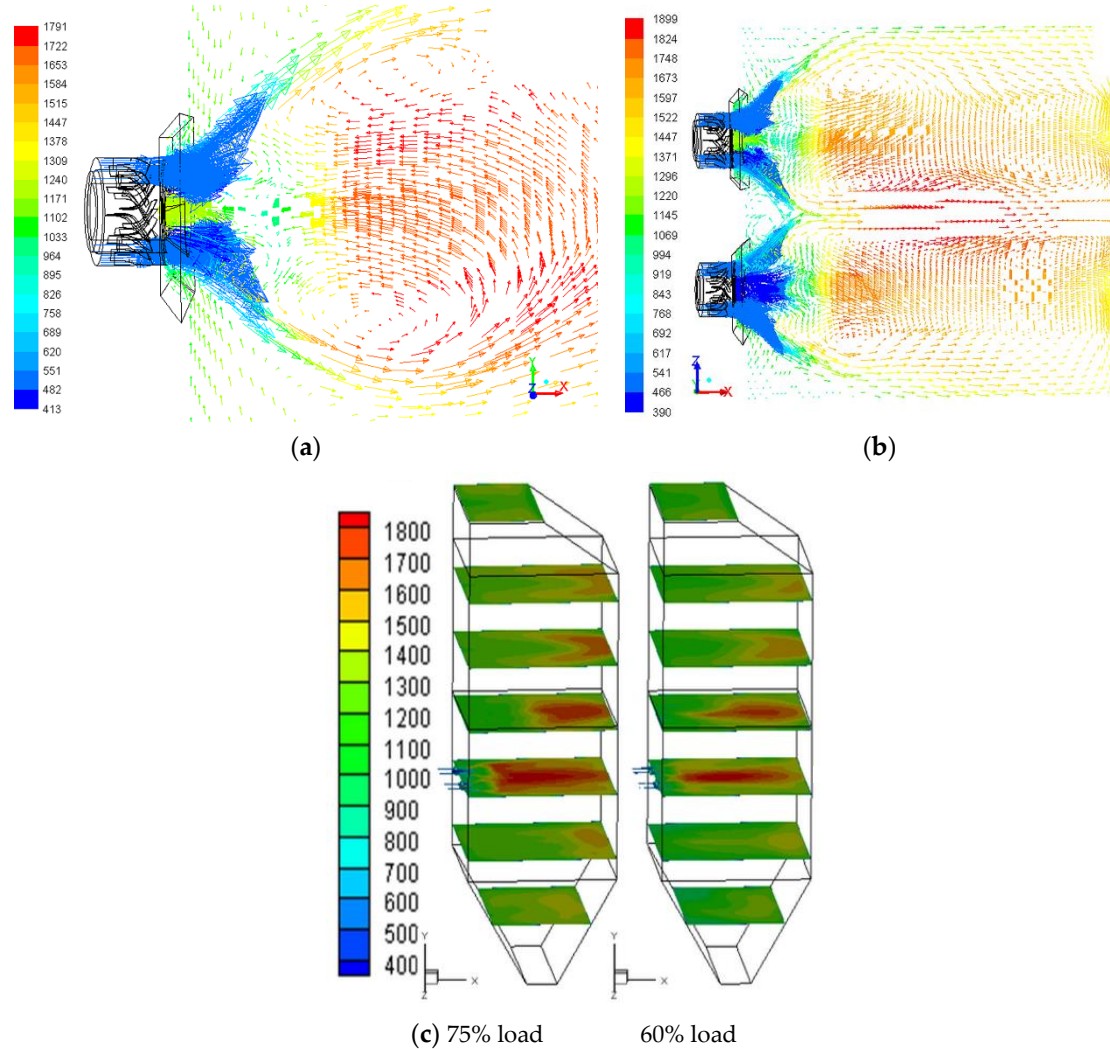

**Figure 5.** Velocity vector colored by temperature in a longitudinal section of the primary zone at z = 0.85 m (**a**) and in the horizontal section y = 4.35 m (**b**), and gas temperature contour on different level sections of the furnace (**c**) with the variation of load (K).

As can be seen from Figure 5, the temperature of the inner and outer SA and CA of the front wall burners were the lowest, with an order of magnitude above 390 K. The geometry of the outer side of the duct end of each burner was a conical structure, which generated reverse pressure action along the jet direction, forming intense eddy currents caused by the entrainment and jetting. The lower temperature mixture of SA near the side walls, and the unburned fuel and combustion products flowed to the exit region of the burner. The eddy currents were conducive to the preheating and stable ignition of pulverized coal particles carried by the PA, forming an ideal flow characteristic of a swirl burner. Then, the gas temperature near the nozzle center increased dramatically. In the middle position near the exit of the burner, two vortexes were small. It can be seen from Figure 5a,b that the fluid of the fuel gas zones of two burners were mixed with each other by jet flow, their flames intermingled and connected together, forming a high temperature zone of intense combustion, with the temperature partially reaching 1800 K. These showed that the heat and mass transfer in the jets were strong, and combustion reactions occur mainly in this area. The flue gas temperature was basically symmetrical on both sides of the wall in the horizontal direction. Figure 5c depicts the temperature characteristics along the level of the furnace. It can be seen that with the variation of the load, the area and length of the high-temperature area of the flame in the furnace were increased, and the temperature was gradually decreased from the primary combustion zone to the exit.

### 4.2. Analysis of Main Combustion Species in the Center Axis of the Burner

Figure 6 shows the variation curves of the molar fractions of the combustion products $O_2$, $CO_2$ and CO and concentration of NO per unit volume in the standard state under two kinds of loads on the sampling line of the central axis of the burner. According to the distribution of the main species, the intensity and characteristics of the reaction between the PA and pulverized coal combustion on the burner axis can be determined.

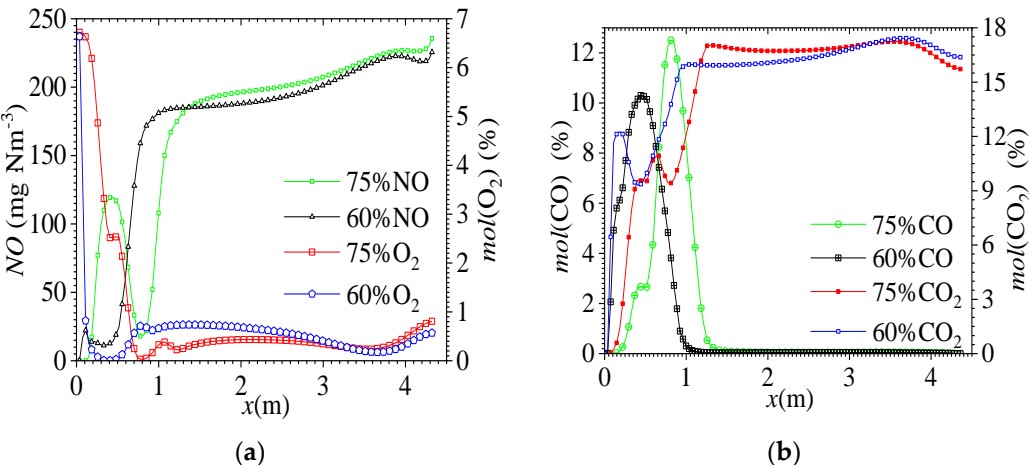

(a)                                      (b)

**Figure 6.** Species variation along the center axis of burner. (**a**) NO concentration and $O_2$ molar fractions; (**b**) CO and $CO_2$ molar fractions.

It can be seen from Figure 6 that the curves can be divided into two regions with obvious characteristics. The critical distance $L$ was taken as the demarcation point of two regions, and $L$ grows longer with an increase in load. In this paper, at 60% load, $L = 1$ m; at 75% load, $L = 1.3$ m. The gas area where $x < L$ was in the diffusion combustion zone and burned under two loads. The molar fraction of $O_2$ decreased sharply with the x-coordinate, reaching a minimum value (Figure 6a). Meanwhile, the molar fraction of CO changed according to an approximate parabola law in this area, and the peak value was higher than 60% of the load at 75% load. At the same time, the molar fraction of $CO_2$ at the corresponding coordinate increased sharply, reaching a first peak, and the $CO_2$ molar fraction generated by combustion at the 60% operating condition was higher than that at 75% load. Then, as the x-coordinate increased, the $CO_2$ concentration firstly decreased and then increased dramatically, reaching a second peak, and the $CO_2$ molar fraction produced by combustion under 60% load was lower than that under 75% load until the furnace outlet front area (Figure 6b).

It was found that at 75% load, the NO concentration had the same characteristics as that of $CO_2$, showing a sharp increase first and then a sharp decrease; the first peak reached 120 mg/$Nm^3$ when $x$ was between 0 and 0.5 m. This is because the temperature of the flue gas increased with the load. Then, the magnitude of the NO concentration increased sharply after a minimum value appeared near 0.8 m. At distance $L$, a sharp increase of 180 g/$Nm^3$ was observed at two loads, with values fluctuating wildly. The NO concentration curve at 60% load was different from in the above conditions, increasing with the depth of the furnace.

It can be seen from Figure 6b that the reducing gas CO's concentration was higher in the species at the corresponding position at 75% load. Because the $O_2$ concentration at the corresponding position was lower, the volatiles from pulverized coal were ignited and burned here. Due to the incomplete combustion, the CO concentration increased. At the same time, a high concentration of CO gas helps to inhibit the NO formation during the combustion of coal char particles [14], resulting in the concentration of NO per unit volume in standard state decreasing slightly with an increase in load. Despite the fact that different combustion reactions and components in the primary combustion zone existed, when $x > L$, the NO concentration gradually increased with an increase in furnace depth.

Meanwhile, the CO molar fraction was very small. The change in the $CO_2$ molar fraction with horizontal distance from the burner and load was not obvious, and the value decreased after $x > 4$ m until the furnace outlet.

### 4.3. Analysis of Flue Gas Temperature and Species NO along the Furnace Height

In order to quantitatively analyze the variation characteristics of the gas temperature and NO concentration along the height direction of the furnace, several horizontal sections were taken in the height direction. The area-weighted average gas temperature and NO concentration under standard state of the flue gas on each horizontal section were calculated, as shown in Figure 7.

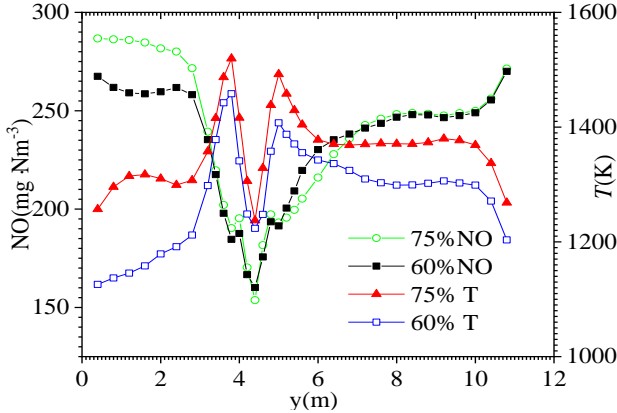

**Figure 7.** Average gas temperature and NO concentration with boiler height.

From Figure 7, it can be seen that the area-weighted average gas temperature near the center cross section of the swirl burner, $y = 4.35$ m, is lower, which is due to the lower temperature of the incoming air and pulverized coal. The heating process and the incomplete combustion process all were carried out around the central axis and height ($y$) direction. With the increase in the swirl radius, the SA was gradually replenished; the mixing of the pulverized coal and air was more and more sufficient. The exothermic reaction of combustion was intense, and the combustion was strongest at the heights of about 3.8 and 5 m, respectively. The flame temperature increased sharply and reached the peak temperature value at two positions, while the temperature gradient near the peak position was very large. This phenomenon was matched with the flow field shown in Figure 5. The lowest gas temperature at 60% load appeared at the bottom of the cold ash bucket; it was different from that at 75% load. In the burn-out zone, the temperature change along the height direction was relatively gentle, while the temperature gradient was large near the 0.8 m area of the furnace top. Most flue gas with high temperature flowed out through the nose region of the furnace.

By contrast, it can be found that the distribution of the gas temperature and NO concentration under the two loads showed opposite change laws partially along the height direction, and the change was the lowest near the level section of the burner axis. This was because the reductive gas CO generated by incomplete combustion reduced most of the NO$x$ that had been generated, and the high concentration of CO was the main factor that inhibited the generation of NO$x$. Taking the height ($y$) at the center axis of the burner as the $Hc$ ($y = 4.35$ m), when $y < Hc$, the concentration of NO decreased sharply with an increase in the y coordinate, and the concentration of NO in the ash bucket was higher at 75% than that at 60% load; when $y > Hc$, the concentration of NO in the swirling combustion zone increased sharply with the rise in temperature but insignificantly with the load; the concentration of NO in the outlet of the furnace was basically the same. In general, the NO concentration was relatively high in the burnout area near the furnace outlet and the lower ash bucket area. In the later stage of the combustion reaction near the furnace outlet, the temperature was higher, the pulverized coal fuel was almost exhausted, and excess air reacted with $NO_2$ to form a large amount of NO$x$. With the

increase in the load, the combustion time and the stroke of pulverized coal particles in the furnace increased obviously. The above analysis mainly studied the NO generation process. Considering that the furnace outlet is the main position for environmental parameter monitoring, the NO, oxygen and carbon burnout rates according to the simulation are listed in Table 4.

**Table 4.** Important parameters of furnace outlet at standard state.

| Load (%) | NO (mg/Nm$^3$) | O$_2$ (%) | Carbon Burn-off Rate (%) |
|---|---|---|---|
| 75 | 226.4 | 6.60 | 99.19 |
| 60 | 219.2 | 6.45 | 99.04 |
| Rate of change (%) | +3.3 | −2.3 | −0.15 |

By comparison, it can be found that at the outlet of the furnace under standard state conditions, the concentration of NO (the emission concentration was slightly higher than 200 mg/Nm$^3$), the volume fraction of O$_2$ and the burnout rate of carbon all decreased with the load reduction. In general, the percentage of change with load did not exceed 5%. In the actual operation process, the air–fuel ratio [6] can be adjusted to meet the requirements of environmental protection. The above study provided basic data for the follow-up energy saving and environmental protection technologies of industrial boilers with similar typical energy consumption equipment.

### 4.4. Comparison of Simulation Results and Experimental Data

When the 35 t/h pulverized coal industrial test boiler was operating under 60% and 75% load conditions, a gas pump thermocouple embedded in the sidewall of the furnace exit region was used to detect the steady-state temperature at $y$ = 10.5 m of the furnace exit temperature region. A total of 12 points were measured in each working condition. The steady-state temperature data of the flue gas were obtained from the position of the measuring point. The simulation results obtained under the same conditions were compared with the measured values. Figure 8 shows a comparison of the predicted temperature field within the boiler and the real-case results for the same position. The simulated and measured temperature values agreed well where $x$ > 0.1 m; the maximum errors of the numerical calculation and measurement values were 5.2% and 2.4%, respectively. An exception was in the boundary zone, where $x$ < 0.08 m, and the results were relatively accurate within the permissible limit. This error was partly due to the assumption of constant particle emissivity [25].

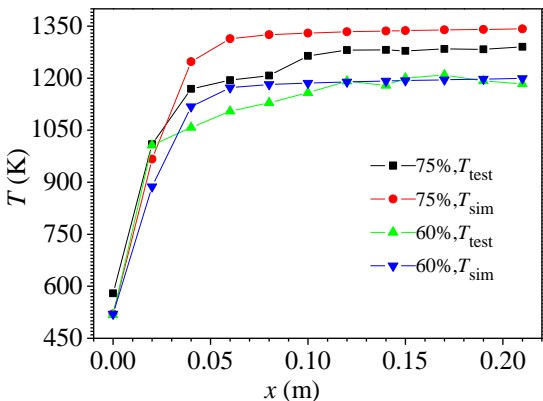

**Figure 8.** Comparison of flue gas temperature between test data and numerical results.

### 5. Conclusions

In this paper, a CFD numerical method was presented to study the combustion performance in 35 t/h industrial pulverized coal boilers with two swirl burners arranged on the wall. The fluid

flow and temperature field of the flue gas near the burner and the distribution characteristics of the combustion products (i.e., NO, $O_2$, CO, $CO_2$ and unburned carbon concentrations) were investigated at two loads to provide basic data for optimizing the combustion operation for nitrogen reduction in industrial boilers. The conclusions were as follows:

1.  The swirl burner of this work can produce an obvious vortex flow region caused by the entrainment, which is conducive to the preheating and stable ignition of pulverized coal particles.
2.  Along the direction of the furnace depth (x) in the burner axis, there was a critical distance $L$; when $x < L$, the $O_2$ molar fraction decreased sharply with the $x$ coordinate. When $x > L$, the NO concentration increased gradually with an increase in furnace depth; the increase value was very small with load. $L$ increased with an increase in load.
3.  In the furnace along the height direction ($y$), take the height of the center axis of the burner as the $H_C$ ($y = 4.35$ m). When $y < H_C$, the NO concentration decreased sharply with an increase in the $y$ coordinate, and the NO concentration at 75% load was greater than that at 60% load in the ash bucket area; when $y > H_C$, the NO concentration increased sharply with the temperature increasing in the swirl combustion zone with no obvious change, and the NO concentration was basically the same at the furnace outlet. The average temperature of the flue gas in the furnace became lower at the $Hc$ position with an increase in the swirl radius of the SA diffuse flow; there were two peak temperatures at two locations.

**Author Contributions:** Y.L., J.H. and Y.G. are the main authors of this manuscript. All the authors contributed to this manuscript. Y.L. and Y.G. conceived the novel idea, and J.H. performed the analysis; L.Z. and Y.M. analyzed the data and contributed analysis tools; L.Z. and Y.L. wrote the entire paper. A.M. and L.Z. checked, reviewed and revised the paper; Y.L. performed final proofreading, and she supervised this research. All authors have read and agreed to the published version of the manuscript.

**Funding:** The authors would like to acknowledge the funding received from the Hi-Tech Research and Development Program of China (2018YFF0216001). The authors also wish to thank Wang L. and Zhang S.S. for their continuous support and useful discussions during the development of the numerical models used in this study.

**Conflicts of Interest:** The authors declare no conflict of interest.

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
