# Peer review of "Numerical Simulation of Combustion in 35 t/h Industrial Pulverized Coal Furnace with Burners Arranged on Front Wall"

_processes, doi:10.3390/pr8101272_

Round 1
Reviewer 1 Report
I have reviewed the paper entitled ‘Numerical Simulation of Combustion in 35t/h Industrial Pulverized Coal Furnace with Burners Arranged on Front Wall’. This manuscript is well presented; graphics and pictures are of good quality. There are however some typos and sentences which will need reviewing.
I believe this manuscript is of good quality. My recommendation is however to accept it with major corrections as there were fundamental aspects of CFD which were not discussed and shown in detail: the results of the mesh independence study, and the justification of the choice of the numerical schemes. If the mesh independence study has been done, this could be included in the manuscript fairly quickly.
The following points will also need to be addressed, in addition to the minor comments highlighted in the annotated copy.
- Abstract:
L21-23: “When x<L… HC (y=4.35).” Sentence to be re-written.
- Introduction:
L65-66: “With respects… conducted studies.” Sentence to be re-written.
- 2 Meshing
This section needs to be improved as the authors mentioned that they refined the boundary layer to meet the y+ requirements, but they did not provide any value or justification and they did not show the results of their mesh-independence study in the following section.
- Mathematical model…
Did the authors used 1st or 2nd order for the equations? Did they modify anything, did they use the default settings? Generally, the choices made by the authors for the CFD settings need to be explained and justified. Why the SIMPLE model, why using standard wall functions?
L172-174: to re-write.
L176 & 177: please remove the path to the Fluent settings to get to the “mark” and “patch” options, this is not of much interest here for readers not familiar with Fluent. Explain what you did rather than providing the full path.
- Section 4.2
L244-245: sentence not clear.

Author Response
Response to Reviewer 1 Comments
peer-review-8267936.(first edition )v1.pdf processes-903279(second)
I have reviewed the paper entitled ‘Numerical Simulation of Combustion in 35t/h Industrial Pulverized Coal Furnace with Burners Arranged on Front Wall’. This manuscript is well presented; graphics and pictures are of good quality. There are however some typos and sentences which will need reviewing.
I believe this manuscript is of good quality. My recommendation is however to accept it with major corrections as there were fundamental aspects of CFD which were not discussed and shown in detail: the results of the mesh independence study, and the justification of the choice of the numerical schemes. If the mesh independence study has been done, this could be included in the manuscript fairly quickly.
Thank you for your valuable comments.
Point 1: My recommendation is however to accept it with major corrections as there were fundamental aspects of CFD which were not discussed and shown in detail: the results of the mesh independence study. If the mesh independence study has been done, this could be included in the manuscript fairly quickly.
Response 1: A good question. We are very appreciated. Thanks for you recommendation. We have studied the theory about Grid Convergence Index (GCI). Ref [17] has been added in the manuscript. According to Ref [17], we have modified as follows:
L121-129:An initial mesh with about 1.40 million nodes cells was first created in the computational domain. The number of mesh nodes cells was then increased to 2.30, 3.0, 4.0 million.The Grid Convergence Index (GCI) was used to quantify the grid independence[17]. The GCI12 for fine and medium grids was 1.21%. The GCI23 for medium and coarse grids was 3.09%. The value of GCI23/(rpGCI12) was 1.015, which was approximately 1 and indicates that the solutions were well within the asymptotic range of convergence.
[17] Roache, P. J. Perspective: A method for uniform reporting of grid refinement studies. J. Fluids Eng1994, 116, 405-413.
Point 2: and the justification of the choice of the numerical schemes.
Response 2: About the choice of the numerical schemes, it has been modified as:
L180-181: The mass, pressure,momentum, chemical species, and energy equations were each discretized using second order upwind scheme .the finite volume approach,
The following points will also need to be addrehssed, in addition to the minor comments highlighted in the annotated copy.
Point 3:Abstract:
L21-23: “When x<L… HC (y=4.35).” Sentence to be re-written.
Response 3: Sentence has been to re-written as :
L20-24: The simulation results showed that there was a critical distance L along the direction of furnace depth(x) and Hc along the direction of furnace height(y) on the burner axis. When x<L, the species varies dramatically.And along the height direction(y): Take the height of the center axis of the burner as the Hc(y = 4.35m). When x<L, the concentration of NO decreases sharply as the height increases, When y<Hc, the NO concentration decreased sharply with the increase of the y coordinate, while increased increasing dramatically with the area-weighted average gas temperature increase in the swirl combustion zone. This study provides a basis for optimizing the operation of nitrogen-reducing combustion and the improvement of burner structure.
Point 4:Introduction:L65-66: “With respects… conducted studies.” Sentence to be re-written.
Response 4: Sentence has been to re-written as :
L68-70: With respects of to reduced mechanisms for the pollutant generation, many researchers have conducted studies.Many researchers have contributed to the study of mechanisms to reduce pollutant production.
Point5: Meshing: This section needs to be improved as the authors mentioned that they refined the boundary layer to meet the y+ requirements, but they did not provide any value or justification and they did not show the results of their mesh-independence study in the following section.
Response 5: To avoid repetition of statement, we have move the related section to the next part(In Mathematical Model and Solution Conditions). See L121-123 and L184-189
The near-wall modeling significantly impacts the fidelity of numerical solutions, inasmuch as walls are the main source of mean velocity and turbulence. Here, standard wall function approach was used for near-wall treatment, also, the local mesh refinement of the boundary layer near the wall surface was performed by adaption to make dimensionless distance y+ meet standard wall function turbulence requirements(y+>15)[19] in globe range. At section y=4.35, y+ value is 50.9, for example.
Point6: Mathematical model…
Did the authors used 1st or 2nd order for the equations? Did they modify anything, did they use the default settings? Generally, the choices made by the authors for the CFD settings need to be explained and justified. Why the SIMPLE model, why using standard wall functions?
Response 6: These sections have been revised.
L180-184:The mass, pressure, momentum, chemical species, and energy equations were each discretized using second order upwind scheme the finite volume approach, the separation and implicit solution scheme based on pressure solver were adopted. In solving algebraic equations, because steady-state calculations generally use SIMPLE algorithm for the pressure correction, the SIMPLE algorithm for the pressure correction it was applied to couple the velocity and pressure fields.
Point7:L172-174: to re-write.L176 & 177: please remove the path to the Fluent settings to get to the “mark” and “patch” options, this is not of much interest here for readers not familiar with Fluent. Explain what you did rather than providing the full path.
Response 7: These sections have been corrected.
L190-200: Before solving the combustion process, the simulation of the cold flow field was conducted, required initialize all regions, set the convergence residual, and the convergence criterion adopted was the RMS (Root Mean Square of the residual values) . and the value adopted was 1 × 10-3 for all equations. After the convergence of the flow field iteration, the hot simulation of models was then started. Energy, radiation, species, and DPM equations was were activated, adapt/region/mark was used to mark the ignition area and local temperature was assigned by patch (Solve/Initialization/Patch Temperature=2000K), and the value of the convergence criterion adopted was 1 × 10-6 for the energy equation. then, iterate all the equations of the models. Number of iterations was set until the solution with successive under-relaxation satisfied the pre-specified tolerance. Some settings and parameters of DPM models in CFD were listed in Table 3.
Table 3. Settings and parameters of DPM models.
|
Non-Premixed Combustion model |
Chemistry State relation |
Chemistry energy treatment |
Chemistry Stream option |
Operating pressure |
PDF Opinion |
|
Settings |
chemistry equilibrium |
Non-adiabatic |
Empirical stream |
101325Pa |
Inlet Diffusion |
|
Discrete Phase model |
DPM Iteration Interval |
Max. Number of Steps
|
Specify Length Scale Length Scale |
Turbulent Dispersion discrete |
Number of tries |
|
Settings |
20 |
5000 |
0.005 |
random walk model |
10 |
Point 8:Section 4.2, L244-245: sentence not clear.
Response 8: This have been corrected.
L264-266: “At the same time, the NO concentration was effectively inhibited, because the existence of a high CO gas concentration was assist in suppressing the high concentration of CO gas helps to inhibit the NO formation during the combustion of coal char particles [14], resulting in that the concentration of NO per unit volume in standard state increased slightly with the increase of load.”
Reviewer 2 Report
Review of the paper (Ref. processes-903279)
“NUMERICAL SIMULATION OF COMBUSTION IN 35T/H INDUSTRIAL PULVERIZED COAL FURNACE WITH BURNERS ARRANGED ON FRONT WALL”
by Jia-de Han, Ling-bo Zhu, Yi-ping Lu, Yu Mu, Azeem Mustafa and Ya-jun Ge
This paper presents research into numerical simulations of combustion process in industrial pulverized coal furnace. The Authors used one of the best methods (Computational Fluid Dynamics) to simulate of the examined device performance.
The strength of presented research is a wide range of simulations (analysis of several different process phenomena) and their possible utilitarian application in environmental protection.
However, I think the Authors could make better use of the post-processing capabilities that Ansys Fluent offers, because of their results are quite limited and predictable.
I have several comments that the Authors could consider when revising the manuscript:
- In my opinion the Authors instead of number of nodes, should rather use number of cells, because this second parameter related to overall dimensions of the device better describes the geometry of the numerical model. In addition, they should add a short information what was a parameter in the mesh-independence criterion - shorter CPU, constancy of predicted quantities, etc. Also a short information whether the quality of the numerical mesh was checked and according to what criterion.
- Part of numerical calculations lacks important information, especially those, concerning solver settings. Similarly - in my opinion - the Authors in the description of the particle distribution, apart from the name of the distribution, should provide its basic parameters adopted during the simulation.
- In the further research, the Authors should further expand the verification part (CFD results vs. measurements, in this work only one quantity - temperature is compared) and - in my opinion - instead of the k-e Realizable turbulence model, check the k-e RNG model with the swirl modification mode. This could improve the consistency of the results.
- The authors of the paper should once again carefully review the text in terms of editing, see e.g. lines: 22,110,120, 139 (it is probably not a Table), 161, 163, 185.
In summary, I think this paper can be published in Processes after revision (with small changes), taking above-mentioned remarks into consideration.

Author Response
Response to Reviewer 2 Comments
Review of the paper (Ref. processes-903279)
“NUMERICAL SIMULATION OF COMBUSTION IN 35T/H INDUSTRIAL PULVERIZED COAL FURNACE WITH BURNERS ARRANGED ON FRONT WALL”
by Jia-de Han, Ling-bo Zhu, Yi-ping Lu, Yu Mu, Azeem Mustafa and Ya-jun Ge
Thank you for your valuable comments. We are appreciated.
Point 1:In my opinion the Authors instead of number of nodes, should rather use number of cells, because this second parameter related to overall dimensions of the device better describes the geometry of the numerical model.
Point 2:In addition, they should add a short information what was a parameter in the mesh-independence criterion - shorter CPU, constancy of predicted quantities, etc. Also a short information whether the quality of the numerical mesh was checked and according to what criterion.
Response 1-2:This have been corrected. L123-130:An initial mesh with about 1.40 million nodes cells was first created in the computational domain. The number of mesh nodes cells was then increased to 2.30, 3.0, 4.0 million. The Grid Convergence Index (GCI) was used to quantify the grid independence[17]. The GCI12 for fine and medium grids was 1.21%. The GCI23 for medium and coarse grids was 3.09%. The value of GCI23/(rpGCI12) was 1.015, which was approximately 1 and indicates that the solutions were well within the asymptotic range of convergence.Mesh-independence test was verified by comparing physical quantities. Finally, the numerical results showed that the number of mesh nodes cells was approximately 2.3 million in Figure 3.
Point 3:Part of numerical calculations lacks important information, especially those, concerning solver settings. Similarly - in my opinion - the Authors in the description of the particle distribution, apart from the name of the distribution, should provide its basic parameters adopted during the simulation.
Response: we have revised. In addition,Table 3 about solver settings has been added.
L180-200: The mass, pressure, momentum, chemical species, and energy equations were each discretized using second order upwind scheme the finite volume approach, the separation and implicit solution scheme based on pressure solver were adopted. In solving algebraic equations, because steady-state calculations generally use SIMPLE algorithm for the pressure correction, the SIMPLE algorithm for the pressure correction it was applied to couple the velocity and pressure fields. The near-wall modeling significantly impacts the fidelity of numerical solutions, inasmuch as walls are the main source of mean velocity and turbulence. Here, standard wall function approach was used for near-wall treatment, also, the local mesh refinement of the boundary layer near the wall surface was performed by adaption to make dimensionless distance y+ meet standard wall function turbulence requirements(y+>15)[19] in globe range. At section y=4.35, y+ value is 50.9, for example.
Before solving the combustion process, the simulation of the cold flow field was conducted, required initialize all regions, set the convergence residual, and the convergence criterion adopted was the RMS (Root Mean Square of the residual values) . and the value adopted was 1 × 10-3 for all equations. After the convergence of the flow field iteration, the hot simulation of models was then started. Energy, radiation, species, and DPM equations was were activated, adapt/region/mark was used to mark the ignition area and local temperature was assigned by patch (Solve/Initialization/Patch Temperature=2000K), and the value of the convergence criterion adopted was 1 × 10-6 for the energy equation. then, iterate all the equations of the models. Number of iterations was set until the solution with successive under-relaxation satisfied the pre-specified tolerance. Some settings and parameters of DPM models in CFD were listed in Table 3.
Table 3.Settings and parameters of DPM models.
|
Non-Premixed Combustion model |
Chemistry State relation |
Chemistry energy treatment |
Chemistry Stream option |
Operating pressure |
PDF Opinion |
|
Settings |
chemistry equilibrium |
Non-adiabatic |
Empirical stream |
101325Pa |
Inlet Diffusion |
|
Discrete Phase model |
DPM Iteration Interval |
Max. Number of Steps
|
Specify Length Scale Length Scale |
Turbulent Dispersion discrete |
Number of tries |
|
Settings |
20 |
5000 |
0.005 |
random walk model |
10 |
Point 4:In the further research, the Authors should further expand the verification part (CFD results vs. measurements, in this work only one quantity - temperature is compared) and - in my opinion - instead of the k-e Realizable turbulence model, check the k-e RNG model with the swirl modification mode. This could improve the consistency of the results.
Response: Thanks for you advice. In the further research, we will try to check the k-e RNG model with the swirl modification mode.
Point 5:The authors of the paper should once again carefully review the text in terms of editing, see e.g. lines: 22,110,120, 139 (it is probably not a Table), 161, 163, 185.
Response: These have been modified. L151:Table 1 title has been deleted.
Answer to round 1comments
Point 11:What’s the combustion equation that is being solved?
Response 11: See lines 139-140: During the combustion of a coal particles, several processes occur which was difficult to model, non-premixed combustion model was used.
Reviewer 3 Report
This is very interesting work demonstrating how parametric changes can result in nitrogen-reduced combustion in industrial pulverized coal burners at different loads. I recommend publication of this paper after minor revisions made as listed below: 1. Elemental composition of the coal is 88.22 wt%. Something is wrong here and numbers should be corrected. 2. Authors should clearly explain why there is increase of NO concentration at 75% load below L =1 m. The CO mole fraction is quite large at L=0.3-0.5 as well as there is significant NO formation in this range. How authors can explain this? 3. It will be very informative if authors can also provide the NO2 concentration and mole fraction at different loadings.Author Response
Comment 1:
Elemental composition of the coal is 88.22 wt%. Something is wrong here and numbers should be corrected.
Response 1:
The values of Table 1 are not the elemental analysis results. They are the ultimate analysis of the coal in dry basis. It is checked that there is no mistake here.
Comment 2:
Authors should clearly explain why there is increase of NO concentration at 75% load below L =1 m. The CO mole fraction is quite large at L=0.3-0.5 as well as there is significant NO formation in this range. How authors can explain this?
Response 2:
In Line 265, we have explained this. The temperature of flue gas increased with the load. It made a increase of NO.
Comment 3:
It will be very informative if authors can also provide the NO2 concentration and mole fraction at different loadings.
Response 3:
Thank you for your advice. The NO2 concentration and mole fraction at different loadings is not in the scope of this paper. We will do this research in further study.
Round 2
Reviewer 1 Report
Many thanks for correcting the manuscript and for addressing most of my initial comments. The manuscript looks now much better.
However, I still have several comments. The first and major point below makes me recommend the manuscript to be accepted, but again with major corrections.
- Mesh independence study. The authors mentioned (Page 4, L126-129) that the Grid Convergence Index (GCI). However, they do not explain much how they obtained the 1.21% and 3.09% values. The reader should be able to understand without having to read reference 17. In addition, these indexes are not enough, the authors should include the curves showing the convergence of the data they extracted from the simulations with the meshes they used, i.e. in this case coarse, medium and fine.
- Choice of algorithms (Page 6 L182-183). Stating that they use SIMPLE because it is usually applied for steady-state simulation does not seem like a good scientific argument. The authors should include a reference showing that this algorithm is usually used for steady-state simulations if this is the case.
- Page 6, L184. “The near wall modelling significantly… as walls are the main source of mean velocity and turbulence”. Can the authors explain this?
- The authors applied y+>15 for wall functions. However, most people consider that y+ should be >30 for wall functions. Can the authors explain this and justify the y+>15 value? A reference would help.
- Page 7, L192. “… the hot simulation of models was then started” This should be re-written.
- Page 7, L198: “… CFD are listed in Table 3”
Author Response
Response to Reviewer 1 Comments
processes-903279(second)
Thanks for comments, and we are appreciated.
Point 1: Mesh independence study. The authors mentioned (Page 4, L126-129) that the Grid Convergence Index (GCI). However, they do not explain much how they obtained the 1.21% and 3.09% values. The reader should be able to understand without having to read reference 17. In addition, these indexes are not enough, the authors should include the curves showing the convergence of the data they extracted from the simulations with the meshes they used, i.e. in this case coarse, medium and fine.
Response 1: We have modified as show in Lines121-126, and Figure 3 which have been added.( in red)
Figure 3. The temperature of monitoring point simulated on three grids and calculated via Richardson extrapolation.
Point 2:Choice of algorithms (Page 6 L182-183). Stating that they use SIMPLE because it is usually applied for steady-state simulation does not seem like a good scientific argument. The authors should include a reference showing that this algorithm is usually used for steady-state simulations if this is the case.
Response 2:We have modified.Lines:175-177, and Refs [20-21]were added.
[20]Patankar, S.; V. Numerical heat transfer and fluid flow. Washington: Hemisphere Publishing Corporation, 1979, 104-156.
[21]Wang, Q.; Guan, J.; Sun, D.H. A simulation of flow field on furnace for horizontal coal smokeless combustion boiler. Proceedings of the CSEE. 2003, 23, 164-167.
Point 3: Page 6, L184. “The near wall modelling significantly… as walls are the main source of mean velocity and turbulence”. Can the authors explain this?.
Response 3: We explain this as: (Lines177-180)Many important scientific and engineering applications involving turbulent flows require the direct integration of the modeled turbulent equations to a surface boundary. Turbulent flows involving boundary layer separating or complex alterations of the surface transport properties[22].
[22] Sarkar, A.; So, R.M.C. A critical evaluation of near-wall two-equation models against direct numerical simulation data. Int. J. Heat and Fluid Flow 1997, 18, 1997: 197-208.
Point 4:The authors applied y+>15 for wall functions. However, most people consider that y+ should be >30 for wall functions. Can the authors explain this and justify the y+>15 value? A reference would help.
Response 4:We have revised as: line 184: y+>30 that most people consider. (in red)
Point 5: Page 7, L192. “… the hot simulation of models was then started” This should be re-written.
Response 5: We have deleted this sentence, revised as lines 188-189: the hot simulation of models was then started. Energy, radiation, species, and DPM equations were then activated. (in red)
Point 6:Page 7, L198: “… CFD are listed in Table 3”
Response 6: We have revised.Lines 191-193:Some settings and parameters of Non-Premixed combustion and DPM models in CFD were listed in Table 3.And the title and format of Table 3 have been modified as follows:
Table 3.Settings and parameters of NPC and DPM models.
|
Non-Premixed Combustion model |
Chemistry State relation |
Chemistry energy treatment |
Chemistry Stream option |
Operating pressure |
PDF Opinion |
|
Settings |
chemistry equilibrium |
Non-adiabatic |
Empirical stream |
101325Pa |
Inlet Diffusion |
|
Discrete Phase model |
DPM Iteration Interval |
Max. Number of Steps
|
Specify Length Scale Length Scale |
Turbulent Dispersion discrete |
Number of tries |
|
Settings |
20 |
5000 |
0.005 |
random walk model |
10 |

Round 3
Reviewer 1 Report
Many thanks for addressing my latest comments. Please check carefully the text as there as few remaining typos, especially in the recently added information.
Author Response
Dear reviewer:
Thank you for your valuable comments. The text of this paper is checked carefully.
This manuscript is a resubmission of an earlier submission. The following is a list of the peer review reports and author responses from that submission.
Round 1
Reviewer 1 Report
The authors studied in this paper a new approach to simulate Pulverized coal combustion in Furnace systems operating in air enrichment (Air Excess) conditions, and using two new swirl burners on the front wall. Modelling a Pulverized coal combustion is a complex process that requires the combination of several models and techniques, and developing a reliable and efficient simulation model is still an important challenge. Numerous strategies have been proposed by studies and the authors cited some overviews on the modelling of Pulverized coal combustion. This new approach fast and stable simulation with reasonably high accuracy and without great computational effort is very interesting using an appropriate numerical method and the validation experimental setup; and the results are very interesting.
The current work has high academic values and I am sure that the new numerical methodology to simulate the process of combustion and will give us in the future a good CFD tool to evaluate CO, H2O, NOx and CO2 for better pulverized coal burner design. Therefore, I would suggest the acceptance of the manuscript, with a few specific comments listed below.
The author does a good turn of literature expanded and the conclusion is very interesting.
- have you tried LES model?
- Have you evaluated the equivalence ratio and air excess from a global equation?
Reviewer 2 Report
The paper lacks being precise. Objectives of the paper are not clear. The paper is very vague. Examples (just from the abstract):
- Line 13: What’s the definition of “an integrated physical model”? This is not clear
- Line 15: What does “efficient” mean? This is not quantified.
- Line 16: CFD is not a principle
- Line 17: What is the capacity of these combustion systems?
The paper requires English editing. Examples (just from the abstract):
- Line 13: “the burners”, does it mean all burners in the market. Why does this line mention “burners” and not “furnaces”?
- Line 17: What does “et al.” mean
The paper lacks being specific.
- It is not clear what are the objectives, research questions, expected outcomes, …?
- How did you quantify the grid independency? Did you use the Grid Convergence Index (GCI)?
- What’s the domain size?
- Assumptions for the boundary conditions are not clear.
- What are the values for the convergence criteria (it mentioned a pre-specified tolerance)?
- What’s the range of y+ values?
- What’s the combustion equation that is being solved?
Separate the results and discussions section.
Figure 3: It seems the number of mesh sizes are not sufficient.
Results are not explained at all. The format of the results section does not walk the reader or reviewer through the novelty of this work.
The paper requires validation and verification of the results with experimental measurements or other studies.
Conclusion section:
- Avoid writing numbered conclusions
- Provide additional details about the results.
Also, the existing knowledge gaps are not identified. The literature review is not focused and does not cover significant studies. Only limited number of relevant studies are cited. Make sure to clearly provide the objectives of this study. The methods require a flowchart or steps to elaborate the tasks of this study.
Reviewer 3 Report
Dear Authors,
Below you will find the article review:
- The quality of Figure 1b could be improved.
- Line 138. Missing last letter in the word "range".
- The quality of Figure 3a could be improved.
Kind regards,
Reviewer